# TCR transgenic clone selection guided by immune receptor analysis and single-cell RNA expression of polyclonal responders

**Nincy Debeuf**[1,2†], **Sahine Lameire**[1,2†], **Manon Vanheerswynghels**[1,2], **Julie Deckers**[1,2], **Caroline De Wolf**[1,2], **Wendy Toussaint**[1,2], **Rein Verbeke**[3], **Kevin Verstaen**[4,5], **Hamida Hammad**[1,2], **Stijn Vanhee**[1,2,6*‡], **Bart N Lambrecht**[1,2,7*‡]

[1]Laboratory of Immunoregulation and Mucosal Immunology, VIB Center for Inflammation Research, Ghent, Belgium; [2]Department of Internal Medicine and Pediatrics, Ghent University, Ghent, Belgium; [3]Laboratory of General Biochemistry and Physical Pharmacy, Faculty of Pharmaceutical Sciences, Ghent University, Ghent, Belgium; [4]VIB Single Cell Core, VIB Center, Ghent, Belgium; [5]Department of Applied Mathematics, Computer Science and Statistics, Ghent University, Ghent, Belgium; [6]Department of Head and Skin, Ghent University, Ghent, Belgium; [7]Department of Pulmonary Medicine, Erasmus University Medical Center Rotterdam, Rotterdam, Netherlands

**\*For correspondence:**
stijn.vanhee@ugent.be (SV);
bart.lambrecht@ugent.be (BNL)

†These authors contributed
equally to this work
‡These authors also contributed
equally to this work

## eLife Assessment

The paper illustrates a **valuable** approach to generating TCR transgenic mice specific for known epitopes. **Solid** evidence validates the described pipeline for identification of TCRs from single-cell datasets for the generation of TCR transgenic mice, while obviating the need for generation of T-cell lines and hybridomas.

**Abstract** Since the precursor frequency of naive T cells is extremely low, investigating the early steps of antigen-specific T cell activation is challenging. To overcome this detection problem, adoptive transfer of a cohort of T cells purified from T cell receptor (TCR) transgenic donors has been extensively used but is not readily available for emerging pathogens. Constructing TCR transgenic mice from T cell hybridomas is a labor-intensive and sometimes erratic process, since the best clones are selected based on antigen-induced CD69 upregulation or IL-2 production in vitro, and TCR chains are polymerase chain reaction (PCR)-cloned into expression vectors. Here, we exploited the rapid advances in single-cell sequencing and TCR repertoire analysis to select the best clones without hybridoma selection, and generated CORSET8 mice (**COR**ona **S**pike **E**pitope specific CD8 **T** cell), carrying a TCR specific for the Spike protein of SARS-CoV-2. Implementing newly created DALI software for TCR repertoire analysis in single-cell analysis enabled the rapid selection of the ideal responder CD8 T cell clone, based on antigen reactivity, proliferation, and immunophenotype in vivo. Identified TCR sequences were inserted as synthetic DNA into an expression vector and transgenic CORSET8 donor mice were created. After immunization with Spike/CpG-motifs, mRNA vaccination or SARS-CoV-2 infection, CORSET8 T cells strongly proliferated and showed signs of T cell activation. Thus, a combination of TCR repertoire analysis and scRNA immunophenotyping allowed rapid selection of antigen-specific TCR sequences that can be used to generate TCR transgenic mice.

## Introduction

One of the key features of adaptive immunity is the enormous receptor diversity of lymphocytes, which is estimated to cover over $10^{10}$ unique sequences for T cells alone (*Cho et al., 2020*; *Lythe et al., 2016*). This empowers the adaptive immune system to tackle an infinite number of invaders. Every single clone cannot expand before antigen encounter because of space constraints and therefore circulates at extremely low precursor frequency. This low precursor frequency has impeded the detailed study of early antigen-specific T cell responses in vivo, despite the advent of peptide-MHC (Major Histocompatibility Complex) tetramers that can detect rare antigen-specific cells after whole body enrichment (*Altman and Davis, 2016*; *Chu et al., 2009*; *Dileepan et al., 2021*; *Moon et al., 2009*; *Moon et al., 2007*; *Shin et al., 2023*). To overcome this detection problem, adoptive transfer of a cohort of naive T cells derived from T cell receptor transgenic mice (TCR Tg), which harbour high numbers of a single clonotypic T cell population, has been extensively used. This technique allowed the tracking of clonal activation, expansion, differentiation, and migration to many model antigens and pathogens, even in early phases of the adaptive immune response, which has greatly contributed to answering key questions in T cell biology in vaccinology, infectious models, cancer, autoimmunity, and allergy (*Attridge and Walker, 2014*; *Coquet et al., 2015*; *Kisielow et al., 1988*; *Miura et al., 2020*; *Oxenius et al., 1998*).

Although many TCR transgenic mouse models are now commercially available and are extensively used in many fields, it is sometimes still needed to rapidly generate a new TCR transgenic mouse. This is certainly the case for emerging pathogens like SARS-CoV-2 coronavirus, for which no research tools were readily available in 2019. The generation of TCR Tg mice necessitates the selection of a functionally relevant TCR clone whose TCR Vα- and Vβ-chains are subsequently expressed in a vector that is highly expressed in thymocytes, causing an effective skewing from a polyclonal to a monoclonal repertoire due to allelic exclusion (*Irving et al., 1998*; *Wang et al., 2001*). To obtain clonotypic information, the coding sequence for the TCR α- and β-chains are extracted from antigen-reactive T cell clones. The most often used method to find rare antigen-reactive clones relies on immunization of mice with relevant antigen, and the ex vivo generation of easily expandable immortal T hybridoma cells from splenocytes. Hybridomas are then selected for reactivity to immunodominant peptides, often independent of cellular phenotype or function, but merely based on upregulation of CD69 or production of IL-2 as a sign for antigen sensitivity. From these hybridomas, the coding regions for the Vα- and Vβ-chain can be amplified by PCR and PCR-cloned into a CD4 expression vector (*Vanheerswynghels et al., 2018*).

The successful selection of a TCR clone that can be used for transgenesis and ultimately as a T cell donor to study antiviral responses depends on two key factors: first, the T cell epitope needs to be antigenic, immunogenic, and stable across pathogenic variants. Second, the TCR clone needs sufficient affinity to the epitope presented on an MHC molecule, as this determines the strength and kinetics of the immune response. In other words, affinity of the TCR determines the proliferative advantage of an effector cell or the tendency to form long-term memory (*Kavazović et al., 2018*). Often, the selection process based on hybridoma selection yields multiple epitope-specific clones that upregulate CD69 or IL-2, and only minimal functional parameters are checked before prioritizing one clone to proceed with. This can result in the generation of TCR Tg mice whose T cells do not respond optimally in a biological setting, fail to compete with endogenous polyclonal T cells upon adoptive transfer, or fail to form memory responses (*Bartleson et al., 2020*; *Milam et al., 2018*; *Weber et al., 2012*).

Here, we provide an optimized approach for TCR clone selection that capitalizes on recent technological advances in the single-cell sequencing field that combine analysis of cellular heterogeneity of responding T cells with immune receptor profiling (VDJseq). We have recently developed the interactive DALI software tool, which is an R software package which allows for fast identification and analysis of T and B cell receptor diversity in high-throughput single-cell sequencing data using command line or a graphical user interface (GUI) (*Verstaen et al., 2022*). Owing to the browser-based interactive GUI, immunologists having limited coding experience can effectively analyse these complex datasets. The DALI tool facilitates linking TCR clonotype information to functional properties, immunophenotype and precursor frequency of individual T cells within a polyclonal response to vaccination or infection in vivo. Information like clonotype diversity, clonotype expansion, and differentially expressed genes between clonotypes allows for rationalized TCR sequence selection, and for generation of TCR Tg mice.

As a proof of concept that this strategy is helpful, we generated CD8 TCR Tg mice carrying a TCR specific for a Spike Epitope of the SARS-CoV-2 Coronavirus. We first generated a (**COR**ona **S**pike **E**pitope specific CD**8 T** cell) **CORSET8** line based on traditional T cell hybridoma technology and selection, which eventually yielded a poorly reactive TCR transgenic line. As an alternative, rationalized approach, we applied scRNA and TCR sequencing and subsequent analysis of the T cell clones via the DALI tool. This latter approach allowed us to a priori evaluate key characteristics required for the development of TCR Tg T cells, including cytokine production, cellular phenotype, and physiologically relevant clonal expansion. Importantly, the generation of fully functional TCR transgenic mice took only a fraction of the workload and time in comparison to the classical approach. CORSET8 mice and the resulting transgenic T cells were thoroughly evaluated by ex vivo and in vivo methods following vaccination and SARS-CoV-2 infection. The streamlined method for generating TCR Tg mice, as presented here, offers an attractive approach for future TCR Tg mouse development.

## Results

### Generation of CORSET8 mice based on hybridoma technology

Designing a TCR transgenic mouse starts with the selection of the antigen epitope to which the TCR is addressed using peptide libraries (*Zhuang et al., 2021*) and/or computational scanning combined with a cellular activity assay (*Erez et al., 2023*; *Vandersarren et al., 2017*; *Vanheerswynghels et al., 2018*). In a first approach to generate CORSET8 Tg mice, we used the IEDB database (*Vita et al., 2019*) to predict the most promising MHCI binding Spike epitopes and made use of the classical, well-established hybridoma technology to create the transgenic mice. To induce Spike-reactive T cells, we immunized C57BL/6 mice by intraperitoneal injection of a mixture of Spike protein and CpG adjuvant every 14 days for a total of three immunizations (*Figure 1-II*, *Figure 1—figure supplement 1*). After fusion of these reactive T cells to the BW5147 lymphoma, the obtained T cell hybridoma clones were scored for their IL-2 production upon stimulation with a Spike peptide pool (INITRFQTL, IWLGFIAGL, GNYNYLYRL, VVFLHVTYV, FQFCNDPFL). Compared to medium control, clone 47 had the strongest fold increase in IL-2 (*Figure 1—figure supplement 1A*). This clone also upregulated CD69 and IFN-γ in the presence of the Spike peptide pool (*Figure 1—figure supplement 1B*) or individual Spike peptides. After comparing the clones in three co-culture experiments, clone 47 was prioritized and used for the generation of a TCR Tg mouse. However, CD8 T cells from the newly generated CORSET8 mice did not respond to Spike peptide or Spike protein in a bone marrow-derived dendritic cell (BM-DC) co-culture experiment (*Figure 1—figure supplement 1C*). To check whether CORSET8 mice had transgenic T cells with a functioning TCR at all, we crossed these mice with *Rag1*-deficient mice that are unable to generate polyclonal T cells. We readily detected CD8 T cells even on the *Rag1*$^{-/-}$ background (*Figure 1—figure supplement 1D*), demonstrating that the TCR was expressed and was of high enough affinity to cause positive selection of CD8 T cells, and therefore functional. Due to the lack of responsiveness, we discontinued this mouse line immediately after it was generated, illustrating that the design of a useful TCR Tg mouse model is an erratic process that can be optimized.

### Generation of CORSET8 mice based on single-cell and TCR repertoire analysis

Given the unsuccessful generation of CORSET8 mice using the classical hybridoma approach, we decided to rationalize clone selection (*Figure 1*). To this end, we immunized mice as described in the previous section (*Figure 1-II*). From these mice, we sorted splenic T cells and used an MHC class I tetramer to discriminate Spike- and non-reactive T cells (*Figure 2—figure supplement 1*). The tetramer identifies CD8 T cells directed to a conserved and highly immunogenic epitope (VNFNFNGL) of the SARS-CoV-2 spike protein. The latter has been confirmed by multiple studies (*Carmen et al., 2021*; *Erez et al., 2023*; *Zhuang et al., 2021*). A naive mouse was taken along as a backdrop for the cellular phenotypes. Ultimately, we subjected three samples for single-cell sequencing: Tetramer positive T cells originating from the immunized mice, Tetramer negative T cells originating from the immunized mice, and tetramer negative T cells originating from the naive control mouse. Clustering of these cells revealed different cellular clusters among sorted CD8 T cells (*Figure 2A*). Tetramer+ cells formed a distinct cluster on the uniform manifold approximation and projection (UMAP) reduced dimensional space (*Figure 2B*). The highest differentially expressed genes of the different clusters are

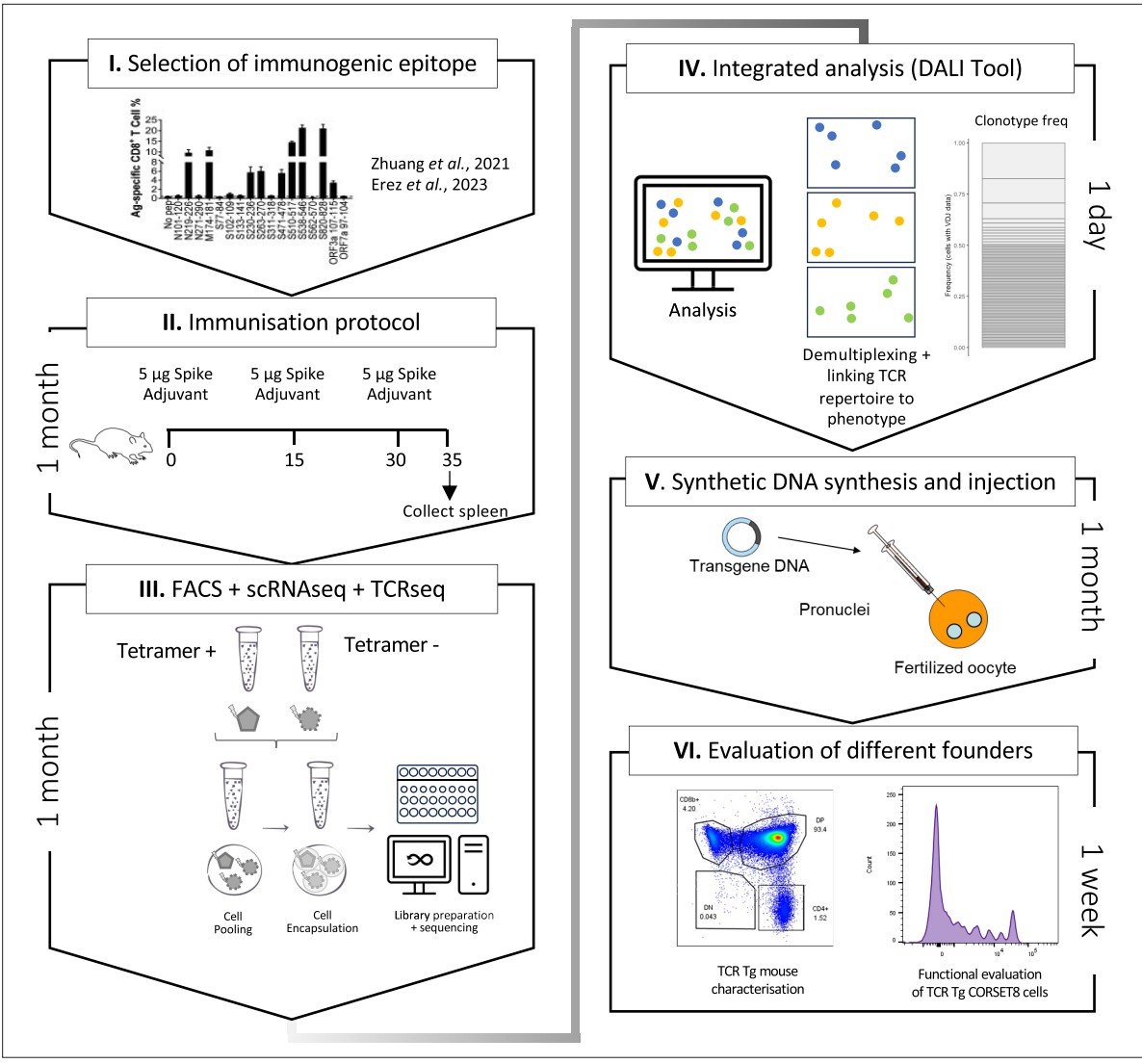

**Figure 1.** Schematic overview of the generation of rationalized CORSET8 transgenic mice. (**I**) Selection of the immunogenic epitope: here based on T cell epitope mapping results of *Zhuang et al., 2021*, which were confirmed by *Erez et al., 2023* (*Erez et al., 2023*; *Zhuang et al., 2021*). (**II**) Immunization protocol to render Spike-reactive T cells (for more information see Material and Methods section). (**III**) Fluorescence-activated cell sorting (FACS) followed by single-cell RNA and T cell receptor (TCR) sequencing. (**IV**) Linking the single-cell RNA and TCR sequencing data allows an integrated analysis to identify clonotypes and compare functional characteristics. (**V**) Injection of the synthetic DNA into a fertilized oocyte followed by transfer into recipient females for gestation, resulting into the birth of transgenic founder offspring. (**VI**) Dual evaluation of the different founders by phenotypical characterization of the transgenic mouse and functional testing of the TCR Tg T cells.

The online version of this article includes the following figure supplement(s) for figure 1:

**Figure supplement 1.** Screening of CD8 T cell hybridoma clones and validation of first generation CORSET8 mice.

plotted in *Figure 2C*. A subgroup of Tetramer+ cells had a distinct phenotype, marked by increased expression of proliferation and activation markers such as *Mki67* and *Gzmb* and transcription factors regulating CD8 effector differentiation such as *Tbet* and *Bhlhe40* (*Li et al., 2019*; *Sullivan et al., 2003*), highlighting their activated T effector state (*Figure 2A–C*). Next, TCRseq data was linked to the Seurat object using our DALI tool (*Verstaen et al., 2022*), which allows for command line or GUI analysis of TCR data (*Figure 2D*). Upon subsetting and reclustering of the Tetramer+ cells from immunized mice, three clusters were defined, of which cluster 1 showed high expression of *Mki67*, while both clusters 0 and 1 showed high expression of *Ifng* and activation markers *Cd69* and *Nr4a1* (*Figure 2E, F*). Assessing clonal expansion, we found that clusters 0 and 1 contained highly expanded clones (*Figure 2G*). When assessing clonal expansion in more detail, we found clonotype1 to be highly

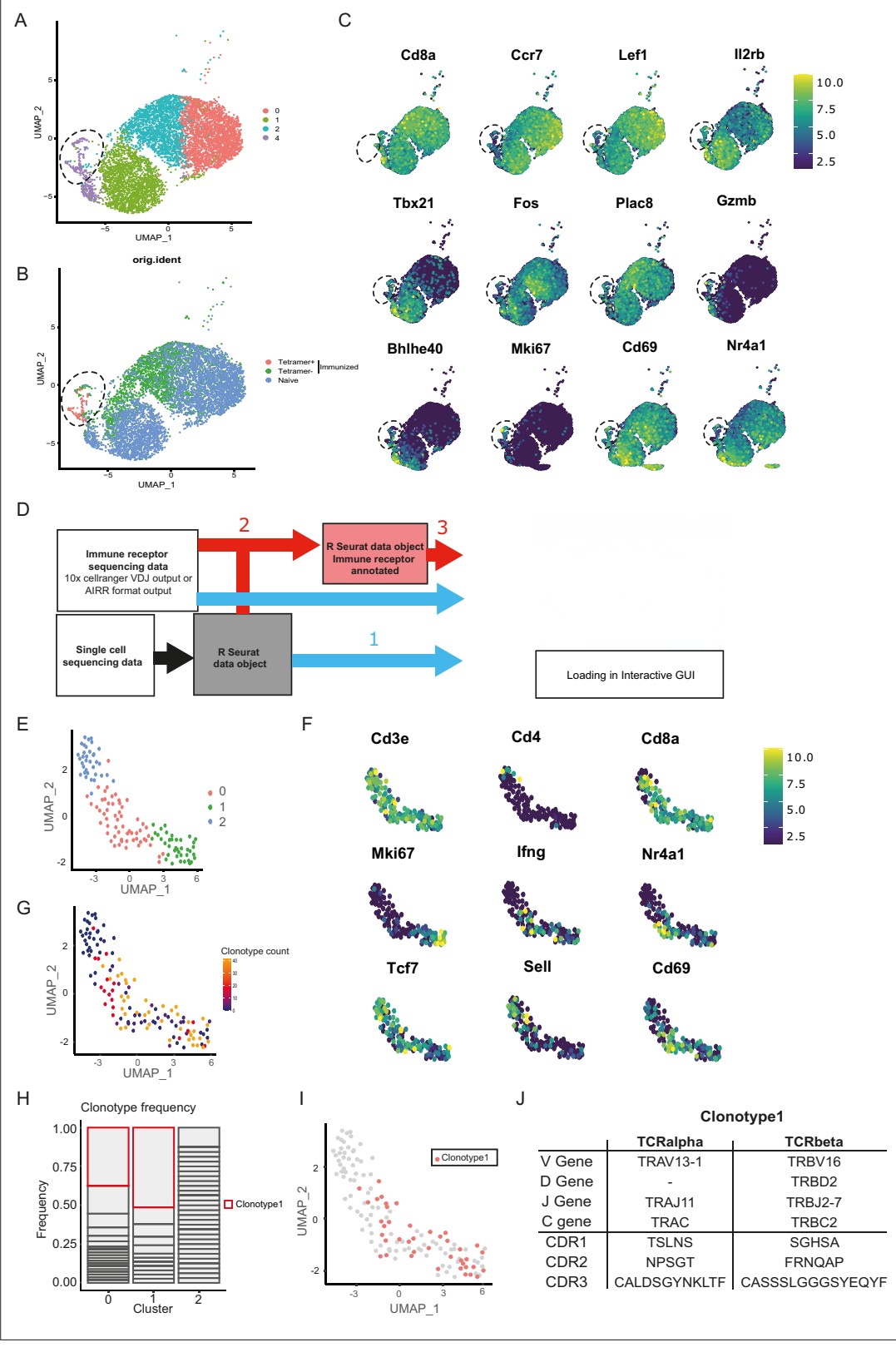

**Figure 2.** Combined single-cell and T cell receptor (TCR) analysis using DALI identified the most promising T cell clone. (**A**) Uniform manifold approximation and projection (UMAP) of splenic CD8 T cell single-cell RNA sequencing visualizing four different clusters based on single-cell analysis. (**B**) Projection of the three sequenced samples on the UMAP: CD8+ Tetramer positive cells of an immunized mouse (sample 1), CD8+ Tetramer negative

*Figure 2 continued on next page*

*Figure 2 continued*

cells of an immunized mice (sample 2) and total CD8 T cells from a naive mouse (sample 3). See *Figure 2—figure supplement 1* for detailed gating strategy of above-mentioned samples. (**C**) Gene RNA expression in splenic CD8 T cells. Hallmark genes among the top differentially expressed genes are depicted. (**D**) Overview of DALI pipelines using either -1- loading of the R Seurat object and immune profiling data directly into the interactive Shiny app or -2- generation of an extended R Seurat object containing immune receptor profiling data, which can be loaded into the interactive Shiny app -3-. (**E**) UMAP of subsetted and reclustered Tetramer+ CD8 T cells showing three different clusters. (**F**) RNA expression in Tetramer+ CD8 T cells of curated activation markers. (**G**) UMAP of Tetramer+ CD8 T cells highlighting clonotype expansion. (**H**) Clonotype frequency in the three different clusters of Tetramer+ CD8 T cells. (**I**) Projection of clonotype 1 on the UMAP of Tetramer+ CD8 T cells. (**J**) TCRα and TCRβ sequence information of clonotype 1.

The online version of this article includes the following figure supplement(s) for figure 2:

**Figure supplement 1.** Gating strategy used to sort Spike Tetramer positive and Tetramer negative splenic CD8 T cells.

expanded in clusters 0 and 1 (*Figure 2H, I*). Given that clonotype 1 expressed proliferation markers (*Mki67$^+$*), produced cytokines (*Ifng$^+$*), had the potential to become a memory cell (*Cd69$^+$*, *Tcf7$^+$*, *Sell$^-$*) and the observed clonal expansion, we decided to continue using this clonotype for the generation of TCR Tg mice (*Figures 1 and 2J*). The selected Vα and Vβ TCR sequences were ordered as gBlocks (IDT) and subsequently cloned into the CD4 expression vector p428 via SalI (*Vanheerswynghels et al., 2018*). Injection fragments we excised from the p428 backbone by NotI digestion, gel purified and injected into fertilized C57BL/6 oocytes.

## CORSET8 mice exhibit skewing to CD8 T cells and TCR Tg cells recognize target peptide ex vivo

In the newly generated (*Rag1*-sufficient) CORSET8 mice, the distribution of splenic CD4 T and CD8 T cells shows a skewing toward CD8 T cells, compared to wildtype mice (*Figure 3A*). Both CD4 T and CD8 T cells displayed normal frequencies of T cell subsets with a significant loss of naive (CD62L$^+$CD44$^-$) and increase of effector (CD62L$^-$CD44$^+$) T cell populations in the CD8 compartment, as expected based on the phenotype of the cells harbouring clonotype 1 (*Figures 3B and 2F*). Staining of the splenocytes with the Spike Tetramer that was initially used to identify target epitope directed T cells, revealed that about 70% of total CD8 T cells of the CORSET mice on a *Rag1* sufficient background were transgenic cells (*Figure 3C*). Next, we aimed to functionally evaluate the generated CORSET8 TCR Tg T cells. Therefore, we co-cultured BM-DCs and Cell Tracer Violet (CTV)-labelled CORSET8 T cells in the presence of increasing doses of Spike peptide (VNFNFNGL, 0–10 µg/ml) for 4 days. The dilution of the cell proliferation dye CTV reflects the augmented proliferation of the CORSET8 cells in correspondence with the antigenic dose (*Figure 3D*). Notably, the CORSET8 cells responded to minute doses of Spike peptide (10 pg/ml), illustrating that the CORSET8 cells express a strong affinity TCR. Furthermore, the CD4 and CD8 T cell distribution in the thymus of the CORSET8 Tg animals shows significant skewing to the CD8+ T cell subset, which also hints toward successful transgene expression and confirms the splenic data in panel 3A (*Figure 3E*).

## CORSET8 cells divide in vivo upon immunization with recombinant spike antigen or with Pfizer BNT162b2 mRNA vaccine

After demonstrating ex vivo CORSET8 T cell proliferation, we aimed to assess this phenomenon in an in vivo context. CORSET8 T cells were isolated and CTV-labelled to track cellular divisions. Next, $1 \times 10^6$ CORSET8 cells were adoptively transferred into C57BL/6 mice that were immunized intratracheally with Spike protein in presence of CpG adjuvant (*Figure 4A*). Four days after T cell transfer, 95.17% (±1.61) of CORSET8 cells underwent division in the draining mediastinal lymph node, as demonstrated by the CTV dilution profile (*Figure 4C*). Both upon Spike/CpG and Pfizer BNT162b2 vaccine immunization, CORSET8 T cells were displaying an activation profile, as shown by increased expression of activation markers such as CD44, IFN-γ, T-Bet, and CD69 compared to endogenous CD45.2 cells (*Figure 4D*). Also Ki-67 expression, as alternative readout than CTV for proliferation, was strongly induced in the CORSET8 T cells. Interestingly, a small population of endogenous CD45.2 cells was Spike Tetramer positive, and these cells had a slightly increased activation profile compared

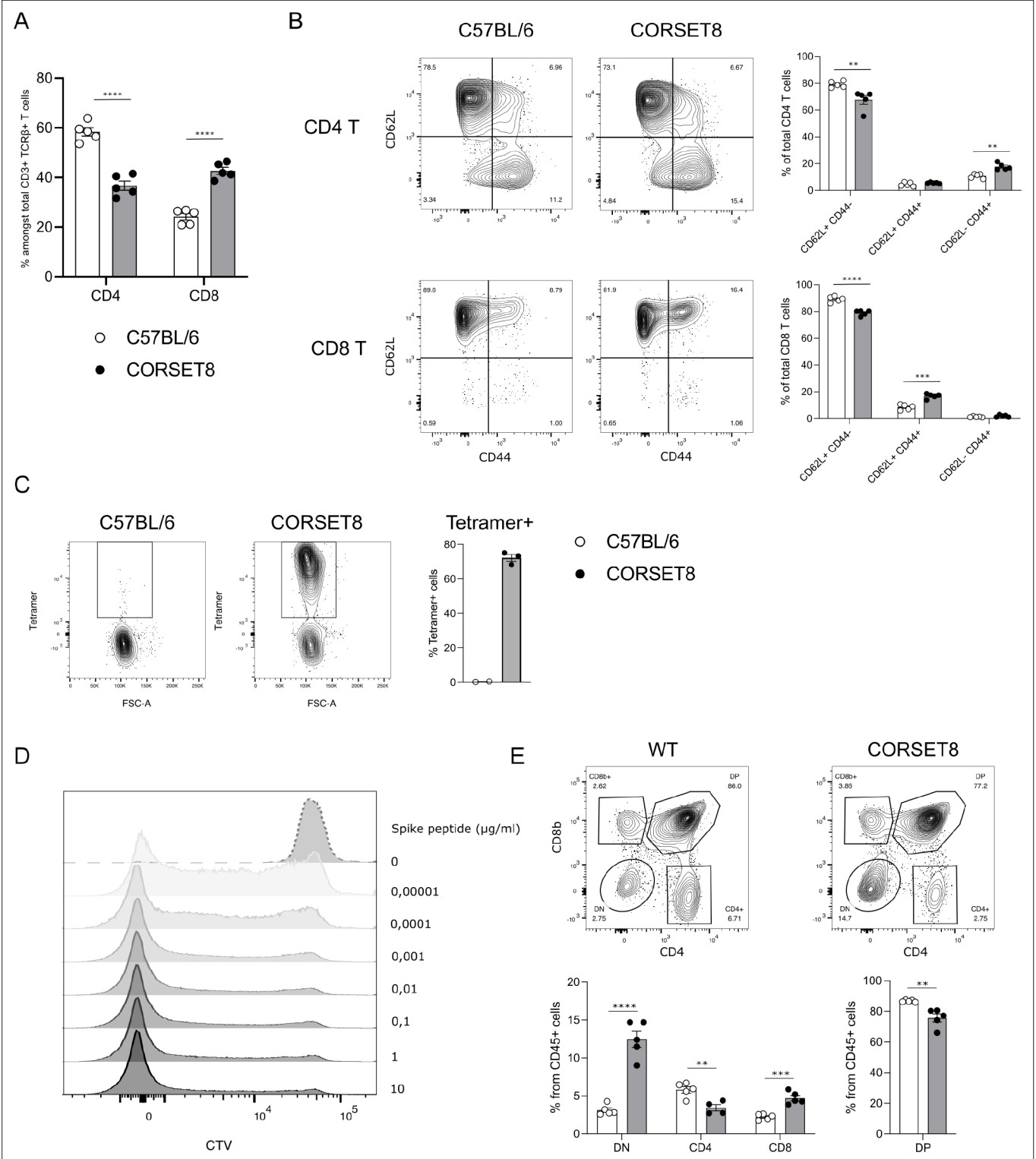

**Figure 3.** CORSET8 mice exhibit normal T cell populations and T cell receptor (TCR) Tg cells recognize target peptide ex vivo. (**A**) Proportion of CD4 T and CD8 T cells among CD3+ TCRβ+ T cells in the spleen of CORSET8 mice and wildtype C57BL/6 littermates. (**B**) Proportion of CD62L+ CD44−, CD62L+ CD44+, and CD62− CD44+ T cells among splenic CD4 T and CD8 T cells. (**C**) Spike Tetramer staining on splenic CD8 T cells of CORSET8 mice or wildtype littermates. (**D**) CTV proliferation profile of CORSET8 T cells in co-culture with bone marrow-derived dendritic cells (BM-DCs) in presence of increasing doses of Spike peptide. (**E**) Visualization of CD4 and CD8 T cells in the thymus of CORSET8 mice and wildtype littermates. Data information: data are shown as means ± SEM. After assessing normality by a Shapiro–Wilk normality test, parametric data were analysed with a $t$-test, whereas non-parametric data were analysed with a Mann–Whitney test. **$p < 0.01$, ***$p < 0.001$, ****$p < 0.0001$. Data are representative of three independent experiments.

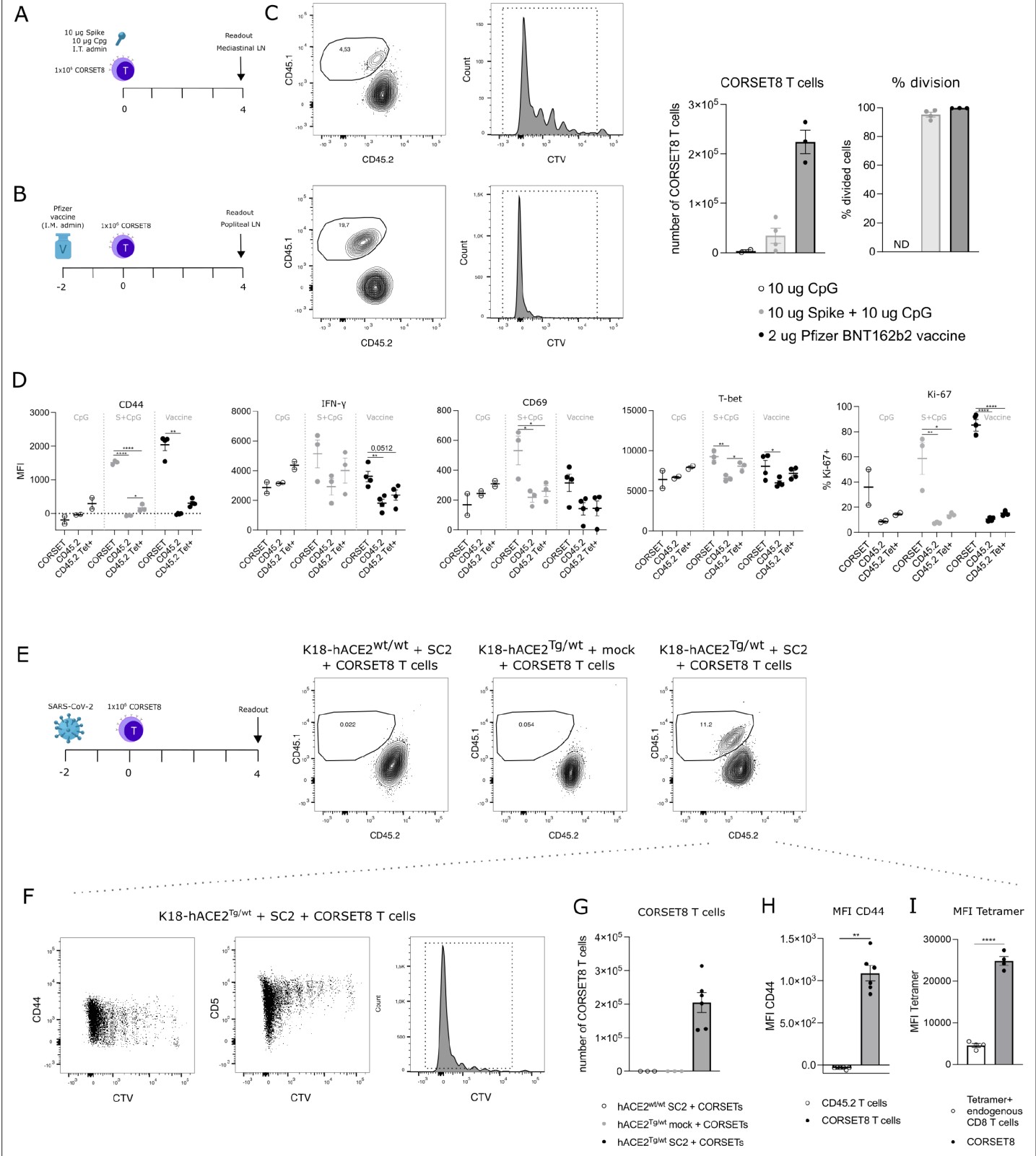

**Figure 4.** CORSET8 cells proliferate in vivo in response to Spike immunization or SARS-CoV-2 infection. (**A, B**) Schematic setup of the experiments. (**C**) CTV dilution profile 4 days after intratracheal immunization with Spike/CpG or intramuscular immunization with Pfizer BNT162b2 vaccine. Quantification of the number of CORSET8 T cells and percentage of divided cells upon immunization with CpG control, Spike/CpG, and Pfizer BNT162b2 vaccine. (**D**) Mean fluorescence intensity (MFI) of several activation markers (CD44, CD69, IFN-γ, T-Bet) and percentage of Ki-67+ cells on both CORSET and

*Figure 4 continued on next page*

*Figure 4 continued*

endogenic CD45.2 T cells after i.t. and i.m. immunizations with Spike/CpG and Pfizer BNT162b2 vaccine, respectively. (**E**) Schematic setup of the experiment and gating on CD45.1$^+$ CORSET8 cells in K18-hACE2$^{wt/wt}$ and K18-hACE2$^{Tg/wt}$ mice upon SARS-CoV-2 or mock infection. (**F**) CTV dilution profile in SARS-CoV-2 infected K18-hACE2$^{Tg/wt}$ mice. (**G**) Number of CORSET8 T cells in K18-hACE2$^{wt/wt}$ and K18-hACE2$^{Tg/wt}$ mice upon SARS-CoV-2 or mock infection. (**H**) MFI of CD44 in CD45.1$^+$ CORSET8 T cells and CD45.2$^+$ T cells of recipient mice. (**I**) MFI of Tetramer staining on CD45.1+CORSET8 T cells and CD45.2+ endogenous Tetramer+ T cells. Data information: data are shown as means ± SEM. In (D), after assessing normality by a Shapiro–Wilk normality test, parametric data were analysed with a one-way ANOVA, whereas non-parametric data were analysed with a Kruskal–Wallis test. In (H), after assessing normality by a Shapiro–Wilk normality test, non-parametric data were analysed with a Mann–Whitney test. *p < 0.05, **p < 0.01, ****p < 0.0001. Data are representative of two independent experiments.

to their Tetramer negative counterparts. However, this increase was still very mild in comparison with the strong activation in CORSET8 T cells. Thus, the phenotypical characteristics for which clonotype 1 was chosen for during single-cell analysis, could be confirmed in the newly generated CORSET8 mice in vivo.

## CORSET8 cells proliferate in response to SARS-CoV-2 infection

As a final validation of our mouse model, we used the CORSET8 cells in an SARS-CoV-2 infection model, conducted under BSL3 conditions (*Figure 4E*). As wildtype mice are not susceptible to SARS-CoV-2 infection, we made use of K18-hACE2 Tg mice, the standard mouse model for SARS-CoV-2 research (*Moreau et al., 2020*; *Rathnasinghe et al., 2020*; *Winkler et al., 2020*; *Yinda et al., 2021*). hACE2$^{Tg/wt}$ mice and their control wildtype littermates were infected with 450 pfu SARS-CoV-2 or mock. Two days after infection, isolated and CTV-labelled CORSET8 T cells were adoptively transferred. Upon analysing the mediastinal lymph nodes 4 days post-adoptive transfer, CORSET8 T cells were exclusively detected in the infected hACE2$^{Tg/wt}$ mice, with no presence observed in mock conditions or hACE2$^{wt/wt}$ infected mice (*Figure 4E, G*). After 4 days, 99.28 ± 0.12% of CORSET8 T wells divided and upregulated activation markers such as CD44 in the draining mediastinal lymph node (*Figure 4F*). Endogenous CD8 T cells from the recipient SARS-CoV-2 infected mice (CD45.2) did not upregulate CD44, demonstrating that only the transgenic T cells were engaged (*Figure 4H*). Furthermore, Spike Tetramer staining revealed that CORSET8 T cells had a higher affinity for the Tetramer than Tetramer+ endogenous CD8 T cells (*Figure 4I*), which again confirms that the generated CORSET8 mice have a high-affinity TCR, a phenotype we selected for.

## Discussion

Adoptive transfer models using TCR transgenic mice as donors of antigen-reactive T cells have made it possible to study early T cell responses at clonal level, which is otherwise very strenuous given the extensive variety of polyclonal T cell clones. This broad variety causes low progenitor frequency of any given clone in the pre-immune repertoire, rendering the tracking of early responses exceedingly difficult. In addition, these models confer a significant advantage to T cell research, as they allow to study the functional status of antigen-specific T cells, but also cell-fate decisions to effector cells, Tfh cells, or memory cells (*Grayson et al., 2000*; *Mueller et al., 2010*; *Tubo et al., 2013*; *West et al., 2011*).

The standard method for generating TCR Tg mice comprises immunization of an animal, isolation of T cells out of lymphoid organs and examination of T cell reactivity by in vitro restimulation of T cell hybridoma clones. TCRs are then identified crudely using PCR primers and antibodies from the antigen-specific T cell clones, often without knowing the detailed sequence of the Vα- and Vβ-chains (*Barnden et al., 1998*; *Bosteels et al., 2020*; *Coquet et al., 2015*; *Nindl et al., 2012*; *Oxenius et al., 1998*; *Plantinga et al., 2013*; *Vanheerswynghels et al., 2018*). The generation of functional TCR Tg mouse models depends on several key steps. A first essential step is the selection of the antigen epitope, which is either done by using peptide libraries (*Zhuang et al., 2021*) or by computational scanning combined with a cellular activity assay (*Erez et al., 2023*; *Vandersarren et al., 2017*; *Vanheerswynghels et al., 2018*). Second, the selected T cell clones need to be able to mount antigen-specific responses. The only way that currently TCRs are screened for this essential characteristic is by stimulating them with antigen peptide or co-culture with antigen-pulsed BM-DCs, whereafter their proliferation, CD69 upregulation and IL-2/IFN-γ cytokine production is assessed (*Bosteels et al., 2020*; *Coquet et al., 2015*; *Nindl et al., 2012*; *Oxenius et al., 1998*). While these assays report

on the potential to activate the TCR bearing clone, it does not inform on fitness/competition of the ensuing clone in a polyclonal environment, nor is it known what the effect of the TCR will be on Tg T cell development into the cell-fate of interest. The importance of the latter has been highlighted by multiple publications. By integrating a single (TCR-transgenic) T cell clone with high- or low-affinity ligands, the impact of affinity on effector and memory potential has been examined (*Knudson et al., 2013*; *Zehn et al., 2009*). In fact, whereas effector cells are mainly found amongst high-affinity clones, memory cells recognize antigens with lower affinity and benefit from more clonal diversity (*Kavazović et al., 2018*). Distinct mechanisms drive effector/memory development in high- and low-affinity T cells, such as regulation of IL-12R signalling, T-bet, and Eomes expression (*Knudson et al., 2013*). It has also been shown that upon weak TCR–ligand interactions, expansion of activated naive T cells will stop sooner than after strong TCR–ligand interactions (*Zehn et al., 2009*). On top of that, tonic signalling also determines cell-fate decisions. For instance, using CD4 TCR-transgenic mice, it has been shown that the strength of tonic signalling determines whether the cells will become Tfh cells or not (*Bartleson et al., 2020*). The same authors showed that the potential to react to tonic MHC signals, reflected by the potential of a clone to upregulate CD5, can also profoundly influence whether a given TCR will induce and immediate effector proliferative response, or rather give a memory response, even when the affinity of two clones for the same immunodominant epitope is nearly identical (*Milam et al., 2018*; *Weber et al., 2012*). The affinity bias of the TCR–MHC interaction leading to long-term memory and residence, particularly in tissues are still not clear. We predict that the method that we propose here which is based on selection of TCRs based on immunophenotype could lead to selection of clones and generation of TCR Tg T cells which are biased to become either immediate effectors or long-term memory cells.

Given the costs and time investment to generate TCR Tg mice, other labs have also invested in optimizing the strenuous process. For example, *Holst et al., 2006* introduced a new model called a retrogenic ('retro' from retrovirus and 'genic' from transgenic) mouse which significantly speeds up the process by using retrovirus-mediated stem cell gene transfer of the TCR alpha and beta sequence (*Holst et al., 2006*). *Guo et al., 2016* described an efficient system for antigen-specific αβTCR cloning and CDR3 substitution (*Guo et al., 2016*). Furthermore, they introduced a novel reporter cell line (Nur77-GFP Jurkat 76 TCRα⁻β⁻) to characterize functional activity of these αβ and γδ TCRs. More recently, a CRISPR-mediated TCR replacement technique was introduced by *Legut et al., 2018* where the endogenous TCRβ sequence gets replaced (*Legut et al., 2018*). In this way, issues like competition with the endogenous TCRs for surface expression or mispairing between endogenous and transgenic TCRs (mixed dimer formation) are excluded. T cell receptor exchange (TRex) mice, where the TCR sequence is integrated in the T cell receptor α constant (TRAC) locus, results in the expression steered by the endogenous promoter and regulatory flanking regions and circumvents the problem of random integration of the construct in the mouse genome. *Rollins et al., 2023* extended the TRex method by disrupting the expression of the gene to which their Tg TCR was directed to, thereby circumventing T cell tolerance (*Rollins et al., 2023*).

In an attempt to further rationalize and streamline the selection of functional T cell clones, with high fitness and phenotype of interest, we combined single-cell gene expression and TCR analysis. To generate our CORSET8 mice, we selected a T cell clone that expressed proliferation markers (*Mki67*⁺), produced cytokines (*Ifng*⁺), and had the potential to become a memory cell (*Cd69*⁺, *Tcf7*⁺, *Sell*⁻). This approach, in which we rationalize every step of TCR Tg mouse generation, can be used to any antigen of interest. Of note, the newly generated DALI tool is user-friendly and free to use, even for immunologists with limited bioinformatics background. We found this approach to be highly time efficient compared to classical methods (see *Figure 1* for timelines), albeit with increased reagent costs for single-cell sequencing. Nonetheless, the rationalized selection of TCR clones might circumvent the expensive generation of poorly functional TCR Tg mouse lines, which additionally conflicts with the reduction arm of the 3R principle.

## Materials and methods
### Mice
Female wildtype C57BL/6 mice were purchased from Janvier Labs (Saint-Berthevin, France) for the immunization experiments. K18-hACE2 transgenic mice were purchased from The Jackson Laboratory

(strain ID 034860). Mice were housed under specific pathogen-free conditions and used between 6 and 12 weeks of age. All experiments were approved by the animal ethics committee at Ghent University (EC2023-097) and were in accordance with Belgian animal protection law.

## Generation of CORSET8 mice via classical hybridoma technology

A detailed description of the protocol can be found in *Vanheerswynghels et al., 2018*. Briefly, C57BL/6 mice were immunized with Spike protein (5 µg, Bio-Techne) in presence of CpG adjuvant (10 µg, ODN1826, Invivogen) according to the immunization protocol in *Figure 1*. Isolated spleen cells were in vitro restimulated with an MHCI peptide pool, based on in silico prediction by the IEDB database. After 4 days of culture, CD8 T cells were positively selected by a CD8 T Mojosort enrichment kit according to the manufacturer's instructions and immortalized by fusion with BW5147 T lymphoma cells. From specificity-selected monoclonal T cell clones (IL2$^+$CD69$^+$ in DC-T cell co-culture), the Vα and Vβ of the CD8+ TCR were subcloned in a p428 expression vector. The TCR constructs were then micro-injected in fertilized oocytes.

## Immunization, infection, and in vivo treatments

For the primary immunizations to generate Spike-reactive T cells, wildtype C57BL/6 were immunized by intraperitoneal injection of 5 µg Spike protein (5 µg, Bio-Techne) in presence of 10 µg CpG adjuvant (10 µg, ODN1826, Invivogen), followed by two reimmunization steps of 5 µg Spike protein in 10 µg CpG adjuvant every 15 days for a total of three immunizations.

For the adoptive transfer experiments, CORSET8 T cells were purified from the TCR Tg donor mice with the MojoSort Mouse CD8 T Cell Isolation Kit (BioLegend) and labelled with Cell Proliferation Dye eFluor 450 (0.01 mM; Thermo Fisher Scientific) for 10 min in the dark at 37°C. Next, $1 \times 10^6$ of these labelled cells (in 100 µl of phosphate-buffered saline [PBS]) were intravenously injected into wildtype or K18-hACE2 Tg C57BL/6 mice. In case of immunizations with Spike protein, C57BL/6 mice were subsequently injected intratracheally with 80 µl containing 10 µg Spike protein and 10 µg CpG as adjuvant. In the vaccine immunization strategy, 2 µg of the COMIRNATY (BNT162b2) vaccine was intramuscularly administered in the thigh muscles of the hind limb (final concentration of 40 µg/ml). To analyse the CORSET8 response to SARS-CoV-2 infection, K18-hACE2 Tg mice were intratracheally infected with a sublethal dose of SARS-CoV-2 virus (450 pfu, SARS-CoV-2 D614G strain SARS-CoV-2/human/FRA/702/2020, obtained from the European Virus Archive (EVAG)). These experiments were performed in Biosafety Level 3 conditions at the VIB-UGent Center for Medical Biotechnology. All intratracheal treatments were given after mice were anesthetized with isoflurane (2 l/min, 2–3%; Abbott Laboratories).

## DC – CD8 T cell co-culture

A detailed description of the generation of BM-DCs can be found in *Vanheerswynghels et al., 2018*. Lymph node cells were harvested from TCR Tg donor mice and CD8+ T cells were purified with MojoSort Mouse CD8 T Cell Isolation Kit (BioLegend) and labelled with Cell Proliferation Dye eFluor 450 (0.01 mM; Thermo Fisher Scientific) for 10 min in the dark at 37°C. 20,000 BMDCs were co-cultured with purified T cells in a 1:5 DC:T cell ratio in sterile tissue culture medium (RPMI (Gibco) containing 10% fetal calf serum (Bodinco), 1.1 mg/ml β-mercaptoethanol (Sigma-Aldrich), 2 mM L-alanyl-L-glutamine dipeptide (Thermo Fisher Scientific) and 56 µg/ml Gentamicin (Thermo Fisher Scientific)) on ice. A serial dilution of antigen was added to the cells, starting from 10 µg/ml of Spike peptide (GNYNYLYRL). CTV dilution and T cell activation were evaluated by flow cytometry after 4 days of incubation at 37°C and 5% $CO_2$.

## Tissue sampling and processing

Mice were euthanized by cervical dislocation or an overdose of pentobarbital (KELA Laboratoria) and organs were collected. To obtain mediastinal lymph node, splenic or thymic single-cell suspensions for flow cytometry, organs were smashed through a 70-µm filter in PBS. Prior to staining for flow cytometry, splenic single-cell suspensions were depleted of red blood cells (RBCs) by using RBC lysis buffer [0.15 M $NH_4Cl$ (Merck), 1 mM $KHCO_3$ (Merck), and 0.1 mM $Na_2$-EDTA (Merck) in MilliQ $H_2O$] produced in-house.

## Flow cytometry and cell sorting

For all flow experiments, single-cell suspensions were incubated with a mix of fluorescently labelled monoclonal antibodies (Ab) in the dark for 30 min at 4°C. To reduce non-specific binding, 2.4G2 Fc receptor Ab (Bioceros) was added. Dead cells were removed from analysis, using fixable viability dye eFluor506 or eFluor780 (Thermo Fisher Scientific). 123count eBeads Counting Beads (Thermo Fisher Scientific) were added to each sample to determine absolute cell numbers. Before acquisition, photomultiplier tube voltages were adjusted to minimize fluorescence spillover. Single-stain controls were prepared with UltraComp eBeads (Thermo Fisher Scientific) following the manufacturer's instructions and were used to calculate a compensation matrix. Sample acquisition was performed on a LSR Fortessa cytometer equipped with FACSDiva software (BD Biosciences) or on a NovoCyte Quanteon flow cytometer (Agilent) equipped with NovoExpress software. Final analysis and graphical output were performed using FlowJo software (Tree Star, Inc) and GraphPad Prism 10 (GraphPad Software, Inc).

The following antibodies were used: anti-CD62L (FITC, eBioscience), anti-Vβ11 (FITC, BD BioSciens, clone RR3-15), anti-CD45.2 (PerCP-Cy5.5, BD Biosciences), anti-TCRβ (BV510, BioLegend), anti-CD45.1 (BV605, BioLegend), anti-CD44 (BV650, BioLegend), anti-CD8 (BV750, BD Biosciences), anti-CD5 (BV786, BD Biosciences), anti-CD44 (RedFluor710, VWR International), anti-CD45 (AF700, BioLegend), anti-CD62L (PE, BD Biosciences), anti-CD8 (PE-Cy7, BioLegend), anti-CD4 (BUV395, BD Biosciences), anti-CD3 (BUV737, BD Biosciences), anti-Ki-67 (BV786, BD Biosciences), anti-CD69 (PE, BD Biosciences), anti-IFN-γ (PE-Cy7, Life Technologies), and anti-T-bet (PE-Cy7, Life Technologies).

## Single-cell RNA seq and downstream DALI analysis

Splenocytes of the immunized mice (and naive control mouse) were stained with an MHCI tetramer recognising our target T cells (H-2K(b) SARS-CoV-2 S 539-546 VNFNFNGL, provided by the NIH Tetramer Core Facility) for 1 hr at 37°C. The cells were washed and stained for surface markers (anti-CD8 (PerCP-eFluor710), Viability dye (eFluor506), anti-CD44 (RedFluor710), and anti-TCRβ (APC-eFluor780)) and a unique TotalSeq-C hashing antibody (BioLegend) for 30 min at 4°C. Afterwards cells were washed and resuspended in PBS + 0.04% bovine serum albumin. Sorted single-cell suspensions were resuspended at an estimated final concentration of 270, 1150, and 1450 cells/μl and loaded on a Chromium GemCode Single Cell Instrument (10x Genomics) to generate single-cell gel beads-in-emulsion (GEM). The scRNA/Feature Barcoding/BCR/TCR libraries were prepared using the GemCode Single Cell 5′ Gel Bead and Library kit, version Next GEM 2 (10x Genomics) according to the manufacturer's instructions. The cDNA content of pre-fragmentation and post-sample index PCR samples was analysed using the 2100 BioAnalyzer (Agilent). Sequencing libraries were loaded on an Illumina NovaSeq flow cell at VIB Nucleomics core with sequencing settings according to the recommendations of 10x Genomics, pooled in a 75:10:10:5 ratio for the gene expression, TCR, BCR, and antibody-derived libraries, respectively.

Demultiplexing of the raw sequencing data was done using bcl2fastq2 (v2.20). Mapping of the gene expression data to the mouse genome (GRCm38; ensembl release 99), barcode processing, unique molecular identifiers filtering, and gene counting was performed using the Cell Ranger suite (version 6.1.1). TCR data were also processed using the Cell Ranger suite, using the cellranger vdj subcommand. The contig annotation was done with the GRCm38 vdj references provided by 10x Genomics (version 5.0.0).

Preprocessing of the data was done by the scran and scater R package according to workflow proposed by the Marioni lab (*Lun et al., 2016*). Outlier cells were identified based on three metrics (library size, number of expressed genes, and mitochondrial proportion) and cells were tagged as outliers when they were five median absolute deviations away from the median value of each metric across all cells. Normalization, detecting highly variable genes, finding clusters, and creating UMAP plots and marker gene identification was done using the Seurat pipeline (v3.1.5). Potential doublets were removed from the analysis. TCR data were integrated into the Seurat object using DALI 2.0.0 for downstream analysis.

## Generation of CORSET8 mice starting from DALI-derived TCR sequences

The selected Vα and Vβ TCR sequences were ordered as gBlocks (IDT) and subsequently cloned into the CD4 expression vector p428 via SalI (*Vanheerswynghels et al., 2018*). Injection fragments we excised from the p428 backbone by NotI digestion, gel purified and injected in an equimolar solution of 1 ng/µl into fertilized C57BL/6 oocytes.

## Statistical analysis

Data information: Data are shown as means ± SEM. All data were analysed with a Shapiro–Wilk normality test to assess whether the data were normally distributed. Parametric data were analysed with a *t*-test or one-way ANOVA test, whereas non-parametric data were analysed with a Mann–Whitney test or Kruskal–Wallis test. *p < 0.05, **p < 0.01, ***p < 0.001, ****p < 0.0001.

## Acknowledgements

We thank the VIB single-cell core, VIB flow core, and VIB transgenic core facility for their expert advice and service in this project. We thank Wendy Toussaint, Leen Vanhoutte, and Tino Hochepied for their help in the generation of TCR Tg mice. We thank Koen Sedeyn for providing training and service in the BSL3 facility at the VIB-UGent Center for Medical Biotechnology. We wish to thank the NIH Tetramer Core Facility for providing H-2K(b) SARS-CoV-2 S 539-546 VNFNFNGL tetramer. The MR1 tetramer technology was developed jointly by Dr. James McCluskey, Dr. Jamie Rossjohn, and Dr. David Fairlie, and the material was produced by the NIH Tetramer Core Facility as permitted to be distributed by the University of Melbourne. This publication was supported by the European Virus Archive GLOBAL (EVA-GLOBAL) project that has received funding from the European Union's Horizon 2020 research and innovation programme under grant agreement No 871029. N D acknowledges support from an FWO Postdoctoral Fellowship – junior grant (1258921N). R V is a postdoctoral fellow of the Research Foundation-Flanders, Belgium (FWO-Vlaanderen; grant No. 1275023N). S V was a postdoctoral fellow of the Research Foundation-Flanders, Belgium (FWO-Vlaanderen; grant No. 1244321N) while working on this project and acknowledges support from FWO Vlaanderen senior research project grant No. G0A7422N. B N L acknowledges support from FWO Methusalem grant (01M01521), FWO EOS research grant (3G0H1222), IBOF grant (01M01521), and the Flanders Institute of Biotechnology (VIB).

## Additional information

### Funding

| Funder | Grant reference number | Author |
|---|---|---|
| Fonds Wetenschappelijk Onderzoek | 1258921N | Nincy Debeuf |
| Fonds Wetenschappelijk Onderzoek | 1275023N | Rein Verbeke |
| Fonds Wetenschappelijk Onderzoek | 1244321N | Stijn Vanhee |
| Fonds Wetenschappelijk Onderzoek | G0A7422N | Stijn Vanhee |
| Fonds Wetenschappelijk Onderzoek | 01M01521 | Bart N Lambrecht |
| Fonds Wetenschappelijk Onderzoek | 3G0H1222 | Bart N Lambrecht |
| iBOF | 01M01521 | Bart N Lambrecht |
| Vlaams Instituut voor Biotechnologie | | Bart N Lambrecht |

| Funder | Grant reference number | Author |
|---|---|---|

The funders had no role in study design, data collection, and interpretation, or the decision to submit the work for publication.

## Author contributions

Nincy Debeuf, Sahine Lameire, Conceptualization, Data curation, Formal analysis, Investigation, Methodology, Supervision, Validation, Visualization, Writing – original draft, Writing – review and editing; Manon Vanheerswynghels, Resources, Supervision, Writing – original draft, Writing – review and editing, Data curation, Validation; Julie Deckers, Conceptualization, Funding acquisition, Resources, Writing – original draft, Data curation, Validation; Caroline De Wolf, Resources, Writing – original draft, Data curation; Wendy Toussaint, Data curation; Rein Verbeke, Visualization; Kevin Verstaen, Methodology; Hamida Hammad, Supervision; Stijn Vanhee, Conceptualization, Data curation, Formal analysis, Funding acquisition, Investigation, Methodology, Software, Supervision, Validation, Visualization, Writing – original draft, Writing – review and editing; Bart N Lambrecht, Conceptualization, Visualization, Supervision, Software, Formal analysis, Investigation

## Author ORCIDs

Nincy Debeuf ⓘ https://orcid.org/0000-0002-4754-4772
Bart N Lambrecht ⓘ https://orcid.org/0000-0003-4376-6834

## Ethics

All experiments conducted in this study were approved by the animal ethics committee at Ghent University (EC2023-097) and were in accordance with Belgian animal protection law.

Reviewer #1 (Public review): https://doi.org/10.7554/eLife.98344.3.sa1
Reviewer #2 (Public review): https://doi.org/10.7554/eLife.98344.3.sa2
Author response https://doi.org/10.7554/eLife.98344.3.sa3

# Additional files

## Supplementary files

MDAR checklist

## Data availability

Sequencing data have been deposited in GEO under accession code GSE284042. The generated mouse line will be made available upon request.

The following dataset was generated:

| Author(s) | Year | Dataset title | Dataset URL | Database and Identifier |
|---|---|---|---|---|
| Debeuf N, Lameire S, Verstaen K, Vanhee S, Lambrecht BN | 2024 | TCR transgenic clone selection guided by immune receptor analysis and single cell RNA expression of polyclonal responders | https://www.ncbi.nlm.nih.gov/geo/query/acc.cgi?acc=GSE284042 | NCBI Gene Expression Omnibus, GSE284042 |

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
