## [Editor Report · eLife Assessment]

The paper illustrates a **valuable** approach to generating TCR transgenic mice specific for known epitopes. **Solid** evidence validates the described pipeline for identification of TCRs from single-cell datasets for the generation of TCR transgenic mice, while obviating the need for generation of T-cell lines and hybridomas.

---

## [Referee Report · Reviewer #1 (Public review)]

Summary:

Debeuf et al. introduce a new, fast method for the selection of suitable T cell clones to generate TCR transgenic mice, a method claimed to outperform traditional hybridoma-based approaches. Clone selection is based on the assessment of the expansion and phenotype of cells specific for a known epitope following immune stimulation. The analysis is facilitated by a new software tool for TCR repertoire and function analysis termed DALI. This work also introduces a potentially invaluable TCR transgenic mouse line specific for SARS-CoV-2.

Strengths:

The newly introduced method proved successful in the quick generation of a TCR transgenic mouse line. Clone selection is based on more comprehensive phenotypical information than traditional methods, providing the opportunity for a more rational T-cell clone selection.

The study provides a software tool for TCR repertoire analysis and its linkage with function.

The findings entail general practical implications in the preclinical study of a potentially very broad range of infectious diseases or vaccination.

A novel SARS-CoV-2 spike-specific TCR transgenic mouse line was generated.

Weaknesses:

The authors present a novel method to develop TCR transgenic mice and overcome the limitations of the more traditional method based on hybridomas.

The authors indicate that they did not intend to directly compare their new method with the traditional hybridoma-based approach. However, such comparison becomes inevitable when the classical method is presented as suboptimal and an alternative approach is introduced to address its limitations. Nevertheless, the explanations provided in their rebuttal have helped clarify their position. The intention behind supplementary figure 1 is to illustrate that a clone that appears suitable using traditional assays may fail to produce a successful TCR transgenic line. This is a valid point that I think should be emphasized more clearly in the manuscript, as it highlights the limitations of the traditional method.

However, the main question that remains is whether the proposed new method will reliably resolve this issue. As previously noted, only one mouse line was generated (successfully) from a single candidate, and the method presented to generate their new TCR transgenic line starts from a more advanced point (a well characterized epitope is already known, and tetramers are available to preselect specific clones). Although this approach likely increases the chances of success, it also limits applicability.

The authors suggest that tetramers are not absolutely necessary to select a clone of interest. Testing this hypothesis would have added value to this manuscript, demonstrating the ability to rapidly generate new TCR transgenic lines in response to emerging pathogens, as outlined in the introduction. They propose that, in such cases, mice could be immunised and expanded clones retested for reactivity. However, it is unclear how this strategy differs from the classic method in increasing the chances of selecting an optimal clone.

Regarding the practical value and cost-effectiveness of extensive expression profiling for T cell clone selection, it remains unclear how well a clone chosen for specific traits will retain these features when developed into a TCR transgenic line, or what traits are ideal for different applications. T cell fate is plastic, and various parameters could influence marker expression.

Issues remain concerning the statistical analysis. Data are said to have been analyzed using both parametric and non-parametric tests. The described approach of performing a normality test followed by either parametric or non-parametric tests is not a correct method for statistical data analysis.

---

## [Referee Report · Reviewer #2 (Public review)]

Summary:

The authors seek to use single-cell sequencing approaches to identify TCRs specific for the SARS CoV2 spike protein, select a candidate TCR for cloning and use it to construct a TCR transgenic mouse. The argument is that this process is less cumbersome than the classical approach, which involves the identification of antigen-reactive T cells in vitro and the construction of T cell hybridomas prior to TCR cloning. TCRs identified by single-cell sequencing that is already paired to transcriptomic data would more rapidly identify TCRs that are likely to contribute to a functional response. The authors successfully identify TCRs that have expanded in response to SARS CoV2 spike protein immunization, bind to MHC tetramers and express genes associated with functional response. They then select a TCR for cloning and construction of a transgenic mouse in order to test the response of resulting T cells in vivo following immunization with spike protein of coronavirus infection.

Strengths:

(1) The study provides proof of principle for the identification and characterization of TCRs based on single-cell sequencing data.

(2) The authors employ a recently developed software tool (DALI) that assists in linking transcriptomic data to individual clones.

(3) The authors successfully generate a TCR transgenic animal derived from the most promising T cell clone (CORSET8) using the TCR sequencing approach.

(4) The authors provide initial evidence that CORSET8 T cells undergo activation and proliferation in vivo in response to immunization or infection.

(5) Procedures are well-described and readily reproducible.

Weaknesses:

(1) The purpose of presenting a failed attempt to generate TCR transgenic mice using a traditional TCR hybridoma method is unclear. The reasons for the failure are uncertain, and the inclusion of this data does not really provide information on the likely success rate of the hybridoma vs single cell approach for TCR identification, as only a single example is provided for either.

(2) There is little information provided regarding the functional differentiation of the CORSET8 T cells following challenge in vivo, including expression of molecules associated with effector function, cytokine production, killing activity and formation of memory. The study would be strengthened by some evidence that CORSET8 T cells are successfully recapitulating the functional features of the endogenous immune response (beyond simply proliferating and expressing CD44). This information is important to evaluate whether the presented sequencing-based identification and selection of TCRs is likely to result in T-cell responses that replicate the criteria for selecting the TCR in the first place.

(3) While I find the argument reasonable that the approach presented here has a lot of likely advantages over traditional approaches for generating TCR transgenic animals, the use of TCR sequencing data to identify TCRs for study in variety of areas, including cancer immunotherapy and autoimmunity, is in broad use. While much of this work opts for alternative methods of TCR expression in primary T cells (i.e. CRISPR or retroviral approaches), the process of generating a TCR transgenic mouse from a cloned TCR is not in itself novel. It would be helpful if the authors could provide a more extensive discussion explaining the novelty of their approach for TCR identification in comparison to other more modern approaches, rather than only hybridoma generation.

Comments on revisions:

The authors have provided additional clarification on the comparisons between the presented method for TCR transgenic generation and the hybridoma method that is more commonly used and added additional verification of the functional response in vivo of T cells expressing the selected TCR. Overall, these additions enhance the evidence that the proposed methods are likely to identify TCRs with a strong immune activation profile and are a reasonable response to the first round of review.

---

## [Author Response]

The following is the authors’ response to the original reviews.

**Public Reviews:**

**Reviewer #1 (Public Review):**
Summary:Debeuf et al. introduce a new, fast method for the selection of suitable T cell clones to generate TCR transgenic mice, a method claimed to outperform traditional hybridoma-based approaches. Clone selection is based on the assessment of the expansion and phenotype of cells specific for a known epitope following immune stimulation. The analysis is facilitated by a new software tool for TCR repertoire and function analysis termed DALI. This work also introduces a potentially invaluable TCR transgenic mouse line specific for SARS-CoV-2.Strengths:The newly introduced method proved successful in the quick generation of a TCR transgenic mouse line. Clone selection is based on more comprehensive phenotypical information than traditional methods, providing the opportunity for a more rational T cell clone selection.The study provides a software tool for TCR repertoire analysis and its linkage with function.The findings entail general practical implications in the preclinical study of a potentially very broad range of infectious diseases or vaccination.A novel SARS-CoV-2 spike-specific TCR transgenic mouse line was generated.Weaknesses:The authors attempt to compare their novel method with a more conventional approach to developing TCR transgenic mice. In this reviewer's opinion, this comparison appears imperfect in several ways:(1) Work presenting the "traditional" method was inadequate to justify the selection of a suitable clone. It is therefore not surprising that it yielded negative results. More evidence would have been necessary to select clone 47 for further development of the TCR transgenic line, especially considering the significant time and investment required to create such a line.

Based on Supplementary Figure 1A only, we understand the concern of the reviewer. However, the data presented in Supplementary Figure 1A is collected during the first rough screening of clones where only the production of IL-2 and IFN-y was measured as a readout for activation. Thereafter, a large selection of responsive clones was further grown and co-cultured with a dose-titration of the antigenic peptide pool. In this second co-culture, also flow cytometry readouts are included such as CD69 expression (as shown in Supplementary Figure 1B). Finally, a narrower selection of responder clones was co-cultured with the different individual peptides to unravel the specificity of the TCR of the clone. In conclusion, the clone was tested at least three times in three distinct set-ups with multiple different readouts.

However, a good evaluation of a clone in an in vitro setting does not necessarily translate in optimal functioning of the cells in a biological context. For instance, some clones survive better in an in vitro setting than others or have already a more activated profile before stimulation.

(2) The comparison is somewhat unfair, because the methods start at different points: while the traditional method was attempted using a pool of peptides whose immunogenicity does not appear to have been established, the new method starts by utilising tetramers to select T cells specific for a well-established epitope.Given the costs and time involved, only a single clone could be tested for either method, intrinsically making a proper comparison unfeasible. Even for their new method, the authors' ability to demonstrate that the selected clone is ideal is limited unless they made different clones with varying profiles to show that a particular profile was superior to others.In my view, there was no absolute need to compare this method with existing ones, as the proposed method holds intrinsic value.

We acknowledge the importance of the well-established hydridoma technology and in no way intended to compare these methods head-to-head, nor do not want to question the validity of the classical methods. The reason why we also wanted to show the failed CORSET8 mouse was to highlight the parts of the TCR generating process which could be rationalized. We again want to emphasize that we do not want to compare methods in any way and recognise that we started from two different bases in terms of clone selection (peptide pool stimulation versus tetramer staining). While the tetramer staining that was employed in the generation of CORSET8 mice allowed to enrich the samples for specific responder clones, this enrichment step is not an absolute requirement for the implementation of the presented method or for the successful generation of a TCR Tg mouse model. An alternative approach could be to use the described method to select for activated and expanded clones upon immunisation and test their reactivity in subsequent steps using peptide stimulation before selecting a receptor. In conclusion, we merely wish to present a novel roadmap for others to use for the generation of their TCR Tg mouse to aid in the selection of the most preferable clone for their purposes.

(3) While having more data to decide on clone selection is certainly beneficial, given the additional cost, it remains unclear whether knowing the expression profiles of different proteins in Figure 2 aids in selecting a candidate. Is a cell expressing more CD69 preferable to a cell expressing less of this marker? Would either have been effective? Are there any transcriptional differences between clonotype 1 and 2 (red colour in Figure 2G) that justify selecting clone 1, or was the decision to select the latter merely based on their different frequency? If all major clones (i.e. by clonotype count) present similar expression profiles, would it have been necessary to know much more about their expression profiles? Would TCR sequencing and an enumeration of clones have sufficed, and been a more cost-effective approach?

The method we present in the paper serves as a proof-of-concept, to be adapted to the researcher’s own needs. We agree with the reviewer that for our intentions with the CORSET8 mice, TCRseq in combination with an enumeration of the clones could also have sufficed and would lower the cost of sequencing. However, we wish to present a roadmap for others to use for the generation of their TCR Tg mouse. Important in this, is that the cellular phenotype, and activation state can be taken into consideration, which might for some projects be essential.

Nonetheless, we do see clear interclonal differences regarding the expression of “activation” genes, where clone 1 is clearly one of the well activated and interferon producing clones (as shown in Author response image 1). As such, researchers could expand these types of analysis to probe for specific phenotypes of characteristics.

(4) Lastly, it appears that several of the experiments presented were conducted only once. This information should have been explicitly stated in the figure legends.

To control for interexperimental variation, every experiment represented in the manuscript has been performed at least two times. We have added the additional information regarding the experimental repetitions and groups in the figure legends.

**Reviewer #2 (Public Review):**
Summary:The authors seek to use single-cell sequencing approaches to identify TCRs specific for the SARS CoV2 spike protein, select a candidate TCR for cloning, and use it to construct a TCR transgenic mouse. The argument is that this process is less cumbersome than the classical approach, which involves the identification of antigen-reactive T cells in vitro and the construction of T cell hybridomas prior to TCR cloning. TCRs identified by single-cell sequencing that are already paired to transcriptomic data would more rapidly identify TCRs that are likely to contribute to a functional response. The authors successfully identify TCRs that have expanded in response to SARS CoV2 spike protein immunization, bind to MHC tetramers, and express genes associated with functional response. They then select a TCR for cloning and construction of a transgenic mouse in order to test the response of resulting T cells in vivo following immunization with spike protein of coronavirus infection.Strengths:(1) The study provides proof of principle for the identification and characterization of TCRs based on single-cell sequencing data.(2) The authors employ a recently developed software tool (DALI) that assists in linking transcriptomic data to individual clones.(3) The authors successfully generate a TCR transgenic animal derived from the most promising T cell clone (CORSET8) using the TCR sequencing approach.(4) The authors provide initial evidence that CORSET8 T cells undergo activation and proliferation in vivo in response to immunization or infection.(5) Procedures are well-described and readily reproducible.Weaknesses:(1) The purpose of presenting a failed attempt to generate TCR transgenic mice using a traditional TCR hybridoma method is unclear. The reasons for the failure are uncertain, and the inclusion of this data does not really provide information on the likely success rate of the hybridoma vs single cell approach for TCR identification, as only a single example is provided for either.

We refer to comments 2 and 3 of reviewer 1 for an answer to this point.

(2) There is little information provided regarding the functional differentiation of the CORSET8 T cells following challenge in vivo, including expression of molecules associated with effector function, cytokine production, killing activity, and formation of memory. The study would be strengthened by some evidence that CORSET8 T cells are successfully recapitulating the functional features of the endogenous immune response (beyond simply proliferating and expressing CD44). This information is important to evaluate whether the presented sequencing-based identification and selection of TCRs is likely to result in T-cell responses that replicate the criteria for selecting the TCR in the first place.

We agree with the reviewer that the data in the initial manuscript included only a limited in vivo functional validation of the CORSET8 T cells. Therefore, we extended these in vivo readouts and measured IFN-g production, CD69, T-bet expression (as measure for activation) and Ki-67 expression (as alternative readout than CTV for proliferation). In the single cell data, we saw that these markers were more pronounced in the selected clone compared to other clones. We could confirm these findings in vivo, and found a stronger induction of IFN-g, CD69, T-bet and Ki-67 in CORSET8 T cells compared to endogenous CD45.2 cells and even Spike-Tetramer+ CD45.2 endogenous cells. We added these data in Figure 4.

(3) While I find the argument reasonable that the approach presented here has a lot of likely advantages over traditional approaches for generating TCR transgenic animals, the use of TCR sequencing data to identify TCRs for study in a variety of areas, including cancer immunotherapy and autoimmunity, is in broad use. While much of this work opts for alternative methods of TCR expression in primary T cells (i.e. CRISPR or retroviral approaches), the process of generating a TCR transgenic mouse from a cloned TCR is not in itself novel. It would be helpful if the authors could provide a more extensive discussion explaining the novelty of their approach for TCR identification in comparison to other more modern approaches, rather than only hybridoma generation.

By integrating the recent technological advances in single cell sequencing into the generation of TCR Tg mice, possibilities arise to rationalize clone selection regarding clonal size, lineage/phenotype and functional characteristics. Often, the selection process based on hybridoma selection yields multiple epitope specific clones that upregulate CD69 or IL-2, and only minimal functional and phenotypic parameters are checked before prioritizing one clone to proceed with. In our experience, transgenic clones selected in this way sometimes render TCR clones unable to compete with endogenous polyclonal T clones in vivo. Taken all these caveats into account, the novelty we present here is that the researcher is fully able to select clones based on several layers of information without the need for extensive or repeated screening. Moreover, the selection of the TCR Tg clone can be done via the interactive and easily interpretable DALI tool. Owing to the browser-based interactive GUI, immunologists having limited coding experience can effectively analyse their complex datasets.

**Recommendations for the authors:**

**Reviewer #1 (Recommendations For The Authors):**
Regarding Supplementary Figure 1A was the experiment conducted more than once? Clone 47 seems minimally superior to the other clones. Incorporating a positive control, such as the response of the OT-I hybridoma to SIINFEKL, could have provided a benchmark to gauge the strength of the observed responses.Also, what was the concentration of the peptide used to restimulate the T cells in vitro? High peptide concentrations can lead to non-specific responses. Ideally, a titration should have been performed, perhaps in a subsequent experiment that only tested those clones that responded well initially. Given the resources required to create and maintain a transgenic mouse line, proceeding with the chosen clone based on the data presented seems to carry considerable risk.

The experiment has been performed three times. The data presented in Supplementary Figure 1A is collected during the first rough screening of clones where only the production of IL-2 and IFN-y was measured as a readout for activation. Thereafter, a large selection of responsive clones was further grown and co-cultured with a dose-titration of the antigenic peptide pool. In this second co-culture, also flow cytometry readouts are included such as CD69 expression (as shown in Supplementary Figure 1B). Finally, a narrower selection of responder clones was co-cultured with the different individual peptides to unravel the specificity of the TCR of the clone. In conclusion, the clone was tested at least three times in three distinct set-ups with multiple different readouts.

In Supplementary Figure 1C, no response to stimulation was detected. Ideally, this figure should have included a positive control, such as PMA/Ionomycin or aCD3/CD28 stimulation.

We agree with the reviewer that this experiment should have included a positive control to validate the non-specific responsiveness of the clone and the technical feasibility of the experiment. Unfortunately, the initial CORSET8 line is frozen and is thus not easily available to repeat the experiment.

Can the authors clarify their gating strategy in the legend of In Supplementary Figure 1D?

Plotted cells are non-debris > single cells > viable cells > CD45+. We have added the information to the legend of Supplementary Figure 1D.

In Figure 2, the figure legend should provide more detail on which cells were sorted for the single-cell RNA sequencing analysis. The materials and methods section explains that cells were stained for CD44. Were activated cells then sorted (either tetramer-positive or -negative), plus naïve CD8 T cells from a naïve mouse?

Supplementary Figure 2 contains the detailed gating strategy during the sort for the single cell experiment. We have added additional red gates to the plots to clarify which samples were sent for sequencing. This has been adapted in the figure legends of both Figure 2 and Supplementary Figure 2.

In Figure 3, Rag1 sufficient transgenic mice display similar numbers of CD4 and CD8 T cells as WT mice in the spleen. Typically, transgenic mice present skewed frequencies of T cells towards the type generated (CD8 in this case), which the authors only found in the thymus of CORSET8 mice. Could this be discussed?

The comment of the reviewer is valid as there is indeed a skewing towards CD8 T cells in the thymi of the CORSET8 mice. We looked back into the data of the experiments and noticed that poor resolution of some markers might have resulted in improper results. We have repeated this and added another T cell marker (TCRbeta) next to the already included CD3e marker. By including both markers, we were able to show that also in spleen the skewing towards the CD8 T cell phenotype is present.

How many repetitions were performed for the experiments in Figures 3D and 3E? How many mice were analyzed for Figure 3E? Please provide this information in the figure legend. Also, include a proper quantification and statistical analysis of the data shown.

New quantification graphs with statistical analysis have been added to Figure 3E. The accompanying figure legend has been adapted. The co-culture displayed in Figure 3D is a representative experiment of two repetitions.

Figure 4C includes 3-4 mice per group. This experiment should have been replicated, and this information should be indicated in the figure legend.

We apologise for omitting this data in the figure legend. The experiment presented in Figure 4A-C has been repeated twice, yielding results following the same trend. We were unable to pool the data as two different proliferation dyes were used in the separate experiments (CFSE and CTV). Furthermore, in the in vivo BSL3 experiments represented in figure 4E-H, we always took along the Spike/CpG-group as positive control. We have added the additional information regarding the experimental repetitions and groups in the figure legend.